# Detection of magnetospheric ion drift patterns at Mars

Chi Zhang [1,2,12], Hans Nilsson[3], Yusuke Ebihara[4], Masatoshi Yamauchi[3], Moa Persson[5], Zhaojin Rong [1,2]✉, Jun Zhong [1,2], Chuanfei Dong [6], Yuxi Chen[6], Xuzhi Zhou [7], Yixin Sun [7], Yuki Harada[8], Jasper Halekas[9], Shaosui Xu[10], Yoshifumi Futaana[3], Zhen Shi[1,2], Chongjing Yuan[1,2], Xiaotong Yun[11], Song Fu[11], Jiawei Gao [1,2], Mats Holmström [3], Yong Wei [1,2]✉ & Stas Barabash[3]

Mars lacks a global magnetic field, and instead possesses small-scale crustal magnetic fields, making its magnetic environment fundamentally different from intrinsic magnetospheres like those of Earth or Saturn. Here we report the discovery of magnetospheric ion drift patterns, typical of intrinsic magnetospheres, at Mars using measurements from Mars Atmosphere and Volatile EvolutioN mission. Specifically, we observe wedge-like dispersion structures of hydrogen ions exhibiting butterfly-shaped distributions (pitch angle peaks at 22.5°–45° and 135°–157.5°) within the Martian crustal fields, a feature previously observed only in planetary-scale intrinsic magnetospheres. These dispersed structures are the results of drift motions that fundamentally resemble those observed in intrinsic magnetospheres. Our findings indicate that the Martian magnetosphere embodies an intermediate case where both the unmagnetized and magnetized ion behaviors could be observed because of the wide range of strengths and spatial scales of the crustal magnetic fields around Mars.

Unlike Earth or Saturn, which possess strong planetary-scale intrinsic magnetospheres, Mars offers a unique magnetic environment in the solar system due to the absence of a planetary-scale intrinsic magnetic field[1,2] and the presence of localized crustal magnetic fields[3,4]. It is widely accepted that the Martian crustal fields have a significant impact on the solar wind interaction with Mars and potentially plays a key role in the planet's atmospheric evolution[5,6]. Numerous studies found that the crustal fields act as magnetic umbrella, shielding the ionosphere from erosion by the solar wind[7–13]. However, contrasting views have been put forth by other researchers, reporting that the

crustal fields could locally enhance the escape rate of ions by facilitating energy transfer between the solar wind and the Martian ionosphere[5,6,14]. To comprehend the role of crustal fields in ion escape, we must address the question, how do the crustal fields control the ion behavior? Answering this question could advance our understanding of the solar wind interaction with Mars and the Martian atmospheric evolution[15–17].

The localized crustal fields are often called mini-magnetospheres because they exhibit morphological similarities to scaled-down intrinsic magnetospheres[18–21]. Nonetheless, due to the differences in

[1]Key Laboratory of Earth and Planetary Physics, Institute of Geology and Geophysics, Chinese Academy of Sciences, Beijing, China. [2]College of Earth and Planetary Sciences, University of Chinese Academy of Sciences, Beijing, China. [3]Swedish Institute of Space Physics, Kiruna, Sweden. [4]Research Institute for Sustainable Humanosphere, Kyoto University, Uji, Japan. [5]Graduate School of Frontier Sciences, The University of Tokyo, Kashiwa, Japan. [6]Center for Space Physics and Department of Astronomy, Boston University, Boston, MA, USA. [7]School of Earth and Space Sciences, Peking University, Beijing, China. [8]Department of Geophysics, Graduate School of Science, Kyoto University, Kyoto, Japan. [9]Department of Physics and Astronomy, University of Iowa, Iowa City, IA, USA. [10]Space Sciences Laboratory, University of California, Berkeley, Berkeley, CA, USA. [11]Department of Space Physics, School of Electronic Information, Wuhan University, Wuhan, China. [12]Present address: Center for Space Physics and Department of Astronomy, Boston University, Boston, MA, USA. ✉e-mail: rongzhaojin@mail.iggcas.ac.cn; weiy@mail.iggcas.ac.cn

strength and spatial scales between the small-scale crustal fields and planetary-scale intrinsic magnetospheres, it remains uncertain whether the crustal fields assume a comparable role to intrinsic magnetospheres in governing plasma (ions and electrons) dynamics. Charged particles within planetary-scale intrinsic magnetospheres tend to become magnetized since they offer a large electromagnetic environment compared with the particle's gyroradius. In this case, the presence of electric fields and inhomogeneity of magnetic field induces particle's drift motion, consequently aiding the formation of drift dispersion structures of magnetized particles[22–28]. Drawing from Earth's magnetosphere as an illustrative example, the magnetic gradient and curvature drift velocity depend on the particles' energy, and give rise to westward motion. Electric drift, however, is independent of energy, resulting in motions to any directions depending on magnetic local times and $L$ ($L$-shell) values. Consequently, distinct spatial distributions of particles at different energies are generated. For example, previous studies have demonstrated that the low-energy portion of injected ions can drift more eastward than the high-energy portion does on the dawnside because of the presence of the convection electric field. The discrepancy is significant at particular $L$[22–27]. As a result, when a spacecraft crosses radially into or out of the inner magnetosphere, it records wedge-like energy dispersion structures in energy versus time spectrograms of ions, characterized by an increase followed by a decrease in ion's energy along the spacecraft trajectory. Hence, these drift dispersion structures commonly manifest as wedge-like energy dispersion patterns in particle's energy versus time spectrograms, which essentially represent distinct spatial distributions of particles at different energies.

Moreover, it is crucial to recognize that the wedge-like dispersion structures of particles require a stable electromagnetic environment where the particle's motions are adiabatic[22,26], otherwise, the chaotic motions of particles would lead to their rapid dissipation. The wedge-like dispersion structures of plasma, are commonly observed within the planetary-scale intrinsic magnetospheres with strong magnetic field strength, such as those of Earth[22–27] and Saturn[28]. Hence, the wedge-like dispersion structures of plasma serve as indicators that the spatial scale of the electromagnetic environment is significantly larger than the plasma's gyroradius.

Recent observations of drift dispersion structures of magnetized electrons within the Martian crustal fields have indicated that the crustal fields exhibit behavior akin to an intrinsic magnetosphere on the scale of the gyroradius of electrons[20]. Consequently, it is expected that the electron dynamics in the crustal fields are similar to those found within intrinsic magnetospheres. In contrast, ions having larger gyroradius are commonly thought to be unmagnetized within the localized Martian crustal fields[29,30], which have weak magnetic field strength and large spatial-temporal inhomogeneity (in contrast to Earth or Saturn's intrinsic dipole fields). Thus, the prospect of observing the magnetized ion behaviors typical for intrinsic magnetospheres (e.g., wedge-like dispersion structures of ions) within the Martian crustal fields is less likely.

Here we present the observations of wedge-like dispersion structures of ions within the Martian crustal fields, based on the high-resolution magnetic fields and plasma measurements obtained by Mars Atmosphere and Volatile EvolutioN mission (MAVEN)[31]. This finding suggests that the magnetized ion behavior commonly observed in intrinsic magnetospheres also occurs at Mars, enhancing our understanding of how the crustal fields control the ion dynamics.

## Results
### Event overview
During 22.50-23.00 UTC, April 15, 2018, MAVEN traversed the nightside terminator region of Mars. The average location of MAVEN was $(-0.62, 0.69, -0.74)$ $R_M$ (radius of Mars, $R_M = 3396$ km) in Mars Solar Orbital (MSO) coordinates. Supplementary Fig. 1 provides a visual representation of the MAVEN spacecraft's path. MAVEN was located above the region with the strongest crustal field, characterized by a longitude of approximately 180°, latitude of approximately 60°S, and an altitude of around 650 km.

An interesting observational feature is that MAVEN detected a series of energy-time dispersed ion signatures (refer to Fig. 1a) with a period of approximately 20-30 seconds during the time interval of 22.52-22.58. MAVEN observed four consecutive rising-tone dispersed structures (energy increase with time), starting at 22.52.32, 22.53, 22.53.25, and 22.53.50 UT (identified by the black dashed curves), with later structures at clearly lower energy than the preceding ones. Later, four falling-tone dispersed structures characterized by energy monotonically decrease with time starting at 22.55.24, 22.55.34, 22.55.48, and 22.56.04 UT were detected. The presence of both rising and falling tones indicates that MAVEN captured spatial dispersion structures of ions rather than a purely temporal effect, which would only produce falling tones[14,30]. These observed features closely resemble the wedge-like dispersion structures commonly observed in Earth's magnetosphere[22–27] and in Saturn's magnetosphere[28].

The energy range of the dispersed ions extends up to approximately 200 eV and down to the lower energy limit of the Solar Wind Ion Analyzer (SWIA) instrument[32], at 25 eV. Figures 1b, c show the energy spectrum of light ions (m/q < 8) and heavy ions (m/q > 12) based on the Suprathermal and Thermal Ion Composition (STATIC) data[33], where it is evident that the energy-dispersed signatures are dependent on the mass of the ions. Specifically, the light hot ions with energies ranging from 20 eV to 200 eV exhibit clear energy dispersion signatures, whereas the heavy ions have relatively low energies (E < 50 eV) and do not show dispersed signatures. The mass-energy spectrum in Fig. 2 indicates that the dispersed ions are primarily hydrogen ions (H$^+$) with m/q = 1, and possibly a small fraction of H$_2^+$ or He$^{2+}$ with m/q = 2. The dominance of H$^+$ applies to all dispersion structures, indicating that these different dispersions are not due to the difference in the mass, which sometimes happens on the Earth[34]. While it is challenging to ascertain how much H$_2^+$ or He$^{2+}$ contributes to the dispersions, here we assume that the dispersed ions are entirely H$^+$. Figure 2 also indicates that heavy ions of Marian origin (O$^+$ and O$_2^+$) do not contribute to the dispersion (the energy is much lower than dispersion structures). Therefore, we suggest that the observed dispersed H$^+$ was likely originated from the solar wind rather than Mars since their energy distribution is distinctly different to that of heavy ions.

The presence of electron voids, characterized by the decrease in suprathermal electron flux (as depicted in Fig. 1d) detected by Solar Wind Electron Analyzer (SWEA)[35], combined with the good agreement between the magnetic fields observed by the Magnetometer (MAG)[36] and the latest spherical harmonic model of the crustal fields[37] (as shown in Fig. 1e), strongly suggests that MAVEN entered a region dominated by crustal fields. However, there are slight differences between the observed magnetic fields and the model fields[37], which may be attributed to the influence of induced fields. Upon scrutinizing it further, it appears that the observed fields can be approximated as a linear superposition of the crustal fields and steady $B_x$-dominated fields (as depicted in Fig. 1f), represented by $\mathbf{B_{obs}} = \mathbf{B_{model}} +$ [14.8, $-0.93$, $-4.15$] nT, where $\mathbf{B_{obs}}, \mathbf{B_{model}}$ correspond to the observed magnetic fields and crustal fields model, respectively. The presence of double-sided loss cone distributions of electrons and the magnetic topology analysis[38] (as shown in Supplementary Fig. 2) further indicates that the field lines primarily consisted of closed crustal field lines with both foot-points located in the nightside ionosphere (MAVEN traversed from about 5 to 3 local time in the early morning sector). Based on these features, we can infer that the observed wedge-like dispersed H$^+$ ions occurred within the mini-magnetosphere formed by the closed field lines of the crustal fields, albeit with additional [14.8, $-0.93$, $-4.15$] nT magnetic fields caused by the influence of induced fields. The observations highlight that the ion drift patterns

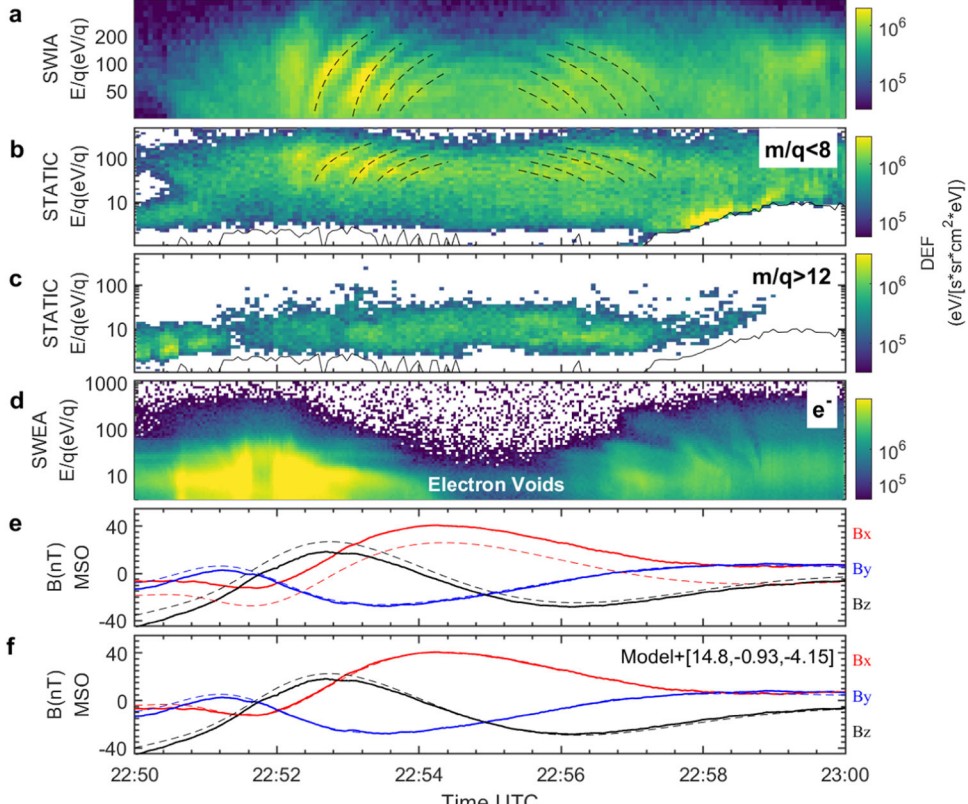

**Fig. 1 | Overview of the wedge-like dispersion structures on April 15, 2018. a** The ions spectrum measured by Solar Wind Ion Analyzer (SWIA) instrument[32]. **b** Energy spectrum of light ions (m/q < 8) measured by the Suprathermal and Thermal Ion Composition (STATIC) data[33]. The black dashed curves were manually fitted to depict the wedge-like dispersion structures. **c** Energy spectrum of heavy ions (m/q > 12) measured by STATIC. The black solid lines in the bottom of panels (b) and (c) are the spacecraft potential. **d** The electron spectrum measured by the Solar Wind Electron Analyzer (SWEA)[35]. The electron voids are characterized by a decrease in suprathermal electron flux. The y-axis and color in panels (a)-(d) are the energy-per-charge (E/q) and the omni-directional differential energy flux (DEF) of particles. **e** The time series of magnetic field measurements in MSO coordinates (refer to Supplementary Fig. 1). The $B_x$, $B_y$, $B_z$ components are represented by red, blue, and black lines, respectively. The solid lines are the magnetic fields measured by Magnetometer (MAG)[36], whereas the dashed lines represent the magnetic fields derived from the crustal fields model[37]. **f** Same as **e**, but the dashed lines represent the superposed fields consisting of the model crustal fields and $B_x$-dominated fields with [14.8, −0.93, −4.15] nT. During this time interval, MAVEN traversed the night-side of the terminator from about 5 to 3 local time (refer to Supplementary Fig. 1).

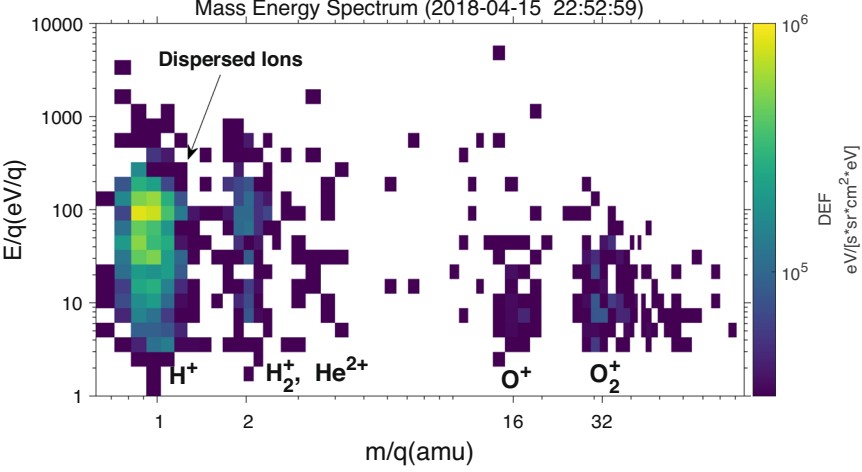

**Fig. 2 | Typical mass-energy spectrum of the dispersed structures.** The dispersed ions with energy ranging from 20-200 eV were $H^+$ with m/q = 1, with a possible small fraction of $H_2^+$ or $He^{2+}$ (m/q = 2). Integrated time is 4 sec. The color represents the DEF of ions.

typically occurred in the intrinsic magnetospheres, the wedge-like dispersion structures, do also occur within the small-scale Martian crustal fields.

## Asymmetric butterfly pitch angle distributions

Figure 3b–d provide the pitch angle distribution of ions in three energy ranges, 30–50 eV, 50–100 eV, and 100–200 eV. It is evident that the dispersed $H^+$ ions across all energy ranges exhibit butterfly-shaped pitch angle distributions[39], characterized by pitch angle peaks at 22.5–45° and 135–157.5°. Upon scrutiny, a notable asymmetry in the butterfly-shaped distribution dictated by the polarity of the radial component of the magnetic fields ($B_r$, refer to Fig. 3e), becomes apparent. Prior to 22.56.09 during which $B_r$ is negative (as indicated by the blue vertical dashed line in Fig. 3), the flux of $H^+$ ions with a pitch angle of 22.5–45° (ions moved toward the planet) is higher than that of ions with a pitch angle of 135–157.5° (ions moved away the planet). Conversely, after 22.56.09 during which $B_r$ is positive, there is a higher flux of $H^+$ ions with a pitch angle of 135–157.5°. These results indicate that there are more inward-moving $H^+$ ions compared to outward-moving ones. The majority of ions with pitch angles of 22.5–45° and 135–157.5° fall within the field-of-view of SWIA (see Supplementary Fig. 3), thereby reinforcing the reliability of the aforementioned results.

## Discussion

The main findings from the observations can be summarized as follows, (1) The dispersed ions are primarily $H^+$ ions, indicating a probable origin from the external solar wind. (2) These periodic dispersed structures exhibit a characteristic period ranging from 10 to 30 seconds. (3) The dispersed ions exhibit asymmetric butterfly-shaped pitch angle distributions. The observation raises several questions. How did the solar wind $H^+$ ions manage to enter the closed field line region created by the crustal fields? What is the cause of the periodicity observed in the dispersed structures? How did the asymmetric butterfly distributions form? Additionally, it is crucial to understand how

these dispersed structures formed. In the subsequent subsections, we will discuss each of these issues in detail.

## Ion injection mechanisms

Two potential mechanisms could explain the injection of solar wind $H^+$. The first involves magnetic reconnection, where solar wind might enter into the crustal fields along open field lines. This scenario seems improbable as magnetic topology analysis reveals no evidence of open field lines (see Supplementary Fig. 2). Yet, we could draw an analogy to the formation of the low-latitude boundary layer observed at Earth[40]. Specifically, the magnetic reconnection taking place at the cusps on both sides could have led to closed field lines and facilitated the injection of solar wind into the crustal fields.

Moreover, the non-adiabatic effect (such as the finite gyroradius effect) represents a second mechanism that could enable the solar wind ions to inject into the crustal fields[29]. Given the constraints of limited observations, it is challenging to demonstrate any of these scenarios.

## Causes of the periodicity

Previous studies have suggested that the bounce motion of particles can also give rise to multiple dispersed structures in Earth's magnetosphere[41]. In such a scenario, the spacecraft would observe dispersed ions alternating between parallel and antiparallel directions as they bounce back and forth[41]. However, our observations reveal that the dispersed structures occur simultaneously in both parallel and antiparallel moving directions (see Supplementary Fig. 4), which contradicts the aforementioned scenario.

The periodic appearance may also originate from upstream plasma waves[14,42]. The upstream waves excited by planetary pickup ions or ions reflected by the bow shock would exhibit a period approximately equal to the proton cyclotron period (on the order of 10 s)[42,43], in alignment with the periodicity of the observed dispersed structures. Thus, we suggest that the waves modulated the heating and

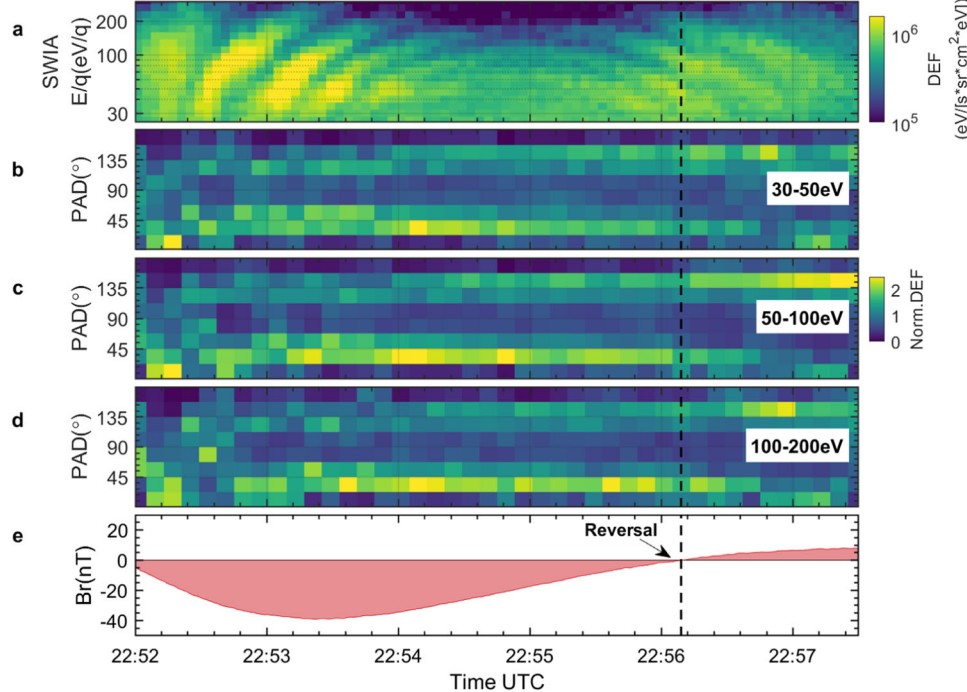

**Fig. 3 | SWIA observations of asymmetric butterfly distribution of the dispersed ions. a** The ions spectrum measured by SWIA. **b–d** show the pitch angle distribution (PAD) normalized by the average differential energy flux (Norm. DEF) at each time between 30–50 eV, 50–100 eV, 100–200 eV, respectively. **e** Time series

of the radial component of magnetic field ($B_r$), where + $B_r$ (-$B_r$) indicates that the field lines are pointing radially outward (inward). The black dashed vertical line denotes the reversal of $B_r$.

reconnection processes, giving rise to the periodic injections, or variations in the source population[22], ultimately leading to the observed periodicity of dispersed ions.

## Mechanisms of asymmetric butterfly distributions

Two potential mechanisms can account for the symmetric butterfly distributions observed in the Earth's inner magnetosphere[44]. In the first mechanism, ions with a pitch angle near 90° or 0°/180° may not be efficiently trapped and are consequently lost, leading to a butterfly distribution of the remaining trapped particles[39]. The loss of ions with pitch angles near 0°/180° could be attributed to the loss cone effect, while the loss of ions with pitch angles near 90° could be attributed to the magnetopause shadowing due to their large gyroradius or drift shell splitting effect due to their drift trajectory[39]. The second possible mechanism is wave-particle interaction[45]. There are no local waves observed (as indicated in Supplementary Fig. 5), but it is possible that heating takes place at low altitudes, and as the heated ions move up into regions with weaker magnetic fields the distributions fold up and form conics due to the mirror force, which could show up as butterfly distributions[46-48]. However, given that the wedge-like distributions appear to be of solar wind origin, we propose that the observed butterfly distributions are primarily caused by the first mechanism, wherein the trapped ions exhibit pitch angles of 22.5–45° or 135–157.5° at the spacecraft's position, while ions with other pitch angles are lost.

Nonetheless, it has difficulty in explaining the asymmetry between 22.5–45° and 135–157.5° pitch angles since the loss cone mechanism predicts nearly symmetric distribution between 22.5–45° and 135–157.5° pitch angles. Thus, the ion velocity distributions at the source might already be anisotropic (lack either parallel or perpendicular ions) when these solar wind $H^+$ accessed the crustal fields, leading to the observations of more inward-moving ions.

## Formation mechanisms of the wedge-like dispersion structures

The wedge-like dispersion structure observed in intrinsic magnetospheres is attributed to the radially spatial separation of ions at different energies[22-27]. To gain a comprehensive understanding of our specific case at Mars, it is essential to investigate the spatial separation of ions at different energies within the crustal fields. A useful approach is to examine the drift path of particles and the spacecraft trajectory relative to the crustal fields on the magnetic equator plane[22-27]. This plane is defined by the magnetic equator points, which correspond to the locations of the minimum field strength ($B_t$) along the field lines. For the Earth's dipole field, the magnetic equator plane is basically parallel to the equator plane and can be easily determined, and we can utilize the $L$-value parameter to quantitatively ascertain the positions of the magnetic field lines. The magnetic equator points situated in lower $L$-shells possess higher $B_t$.

Here we also construct a magnetic equator plane of the crustal fields to determine the spacecraft's relative path and gain insights into the spatial distribution of ions at different energies. First, we extracted the magnetic equator points of the traced field lines (denoted by the magenta dots in Fig. 4a). Subsequently, we employed the least-square method to fit the average magnetic equator plane (depicted by the grey-shaded region) that encompasses most of these magnetic equator points. Figure 4a demonstrates that these magnetic equator points predominantly reside within the magnetic equator plane, indicating that this plane is well-determined. Then, similar to the dipole field, we set a local coordinate system to study the distribution of ions at different energies, where **n** is the normal direction of the magnetic equator plane, **a** is the azimuthal direction, **r** is the radial direction (pointing to the outer region of crustal fields) which completes the right-handed system (see detailed information in the Methods).

The distribution of $B_t$ of the magnetic equator points in the magnetic equator plane is depicted in Fig. 4b. It is evident that the $B_t$ increases as the relative radial distance decreases, indicating that

magnetic equator points with higher $B_t$ correspond to field lines located in the more interior region of the crustal fields, similar to the dipole field pattern. Therefore, although we cannot employ the conventional $L$-value parameter for quantitative determination of magnetic field lines positions as done on Earth, we can qualitatively assess the relative radial positions of the field lines of crustal fields by comparing the $B_t$ of their magnetic equator points. The magnetic equator points of the field lines situated in the more interior region of the crustal fields will have a higher $B_t$.

We also extracted the magnetic equator points of the field lines crossed by MAVEN during this period (marked by blue dots in Fig. 4a), and obtained their $B_t$ value (refer to Fig. 4c). We can find that the $B_t$ of these magnetic equator points initiates at approximately 25 nT, then increases and reaches its maximum value of around 45 nT at 22.54.45, and subsequently gradually decreases to about 33 nT. These variations indicate that MAVEN was progressively entering the interior region of the crustal fields during 22.52.30-22.54.45 (see the blue arrow in Fig. 4b), resulting in an inward movement of about 150 km. Consequently, the estimated radial inward speed of MAVEN within the magnetic equator plane is approximately 1.1 km/s. Subsequently, MAVEN moved towards to the outer part of crustal fields during 22.54.45-22.57.30.

Importantly, each individual dispersed structure (refer to Fig. 4d) reveals a clear trend, as MAVEN moved toward to the inner (outer) part of crustal fields, the energy of the ions increased (decrease). This suggests that the high-energy portion of each individual dispersed structure primarily resides in the inner part of the crustal fields, while low-energy portion tends to occupy the outer part. For instance, for the first rising tone, MAVEN observed a notable increase in ion energy from about 20 eV at 22.52.30 to about 200 eV at 22.53.10. This increase is accompanied by a corresponding rise in the $B_t$ values of the magnetic equator points, which went from approximately 25 nT to 35 nT. This suggests that the 200 eV ions were located approximately 100 km deeper inside the crustal fields compared to the 20 eV ions on the magnetic equator plane (refer to Fig. 4b). Consequently, we infer that, in a single injection scenario, the injected ions drifted toward the center of the crustal fields, with higher energy ions penetrating deeper into the crustal fields.

According to the above results, the complete scenario of observed wedge-like dispersions could be obtained (refer to Fig. 5). Let's start with the case of only one injection (refer to the younger injection represented by violet color in Fig. 5). All injected ions exhibited a drift motion towards the center of the crustal fields. The 200 eV ions (dark violet) could drift deeper into the inner region compared to the 20 eV ions (light violet), leading to the observation of one rising tone (falling tone) when MAVEN moved towards the inner (outer) part of the crustal fields.

The two injections would yield the observations of two rising and falling tones. As depicted in Fig. 5, the red color represents the older injection event that occurred 10-30 s earlier than the younger injection (violet color). Thus, the ions from the older injection experience 10-30 s longer drifting time than that of the younger injection, allowing them to penetrate deeper into the crustal fields compared to the ions from the younger injection. Consequently, the spacecraft could only detect the dispersed ions with energy ranging from 20 to 100 eV for the older injection since the ions with energy higher than 100 eV of the older injection had penetrated deeper than the spacecraft. Thus, it is reasonable to infer that the older (younger) injection exhibited a lower (higher) observed peak energy and occupied a more interior (outer) region of the crustal fields, characterized by a higher (lower) $B_t$ of magnetic equator points. With the knowledge gained from the two injections, we can readily extrapolate the results for four injections. The rising and falling tones observed in the outermost (innermost) region of the crustal fields correspond to the youngest (oldest) injection, which possesses the highest (lowest) observed peak energy. These findings align well with the observations presented in Fig. 4b–d.

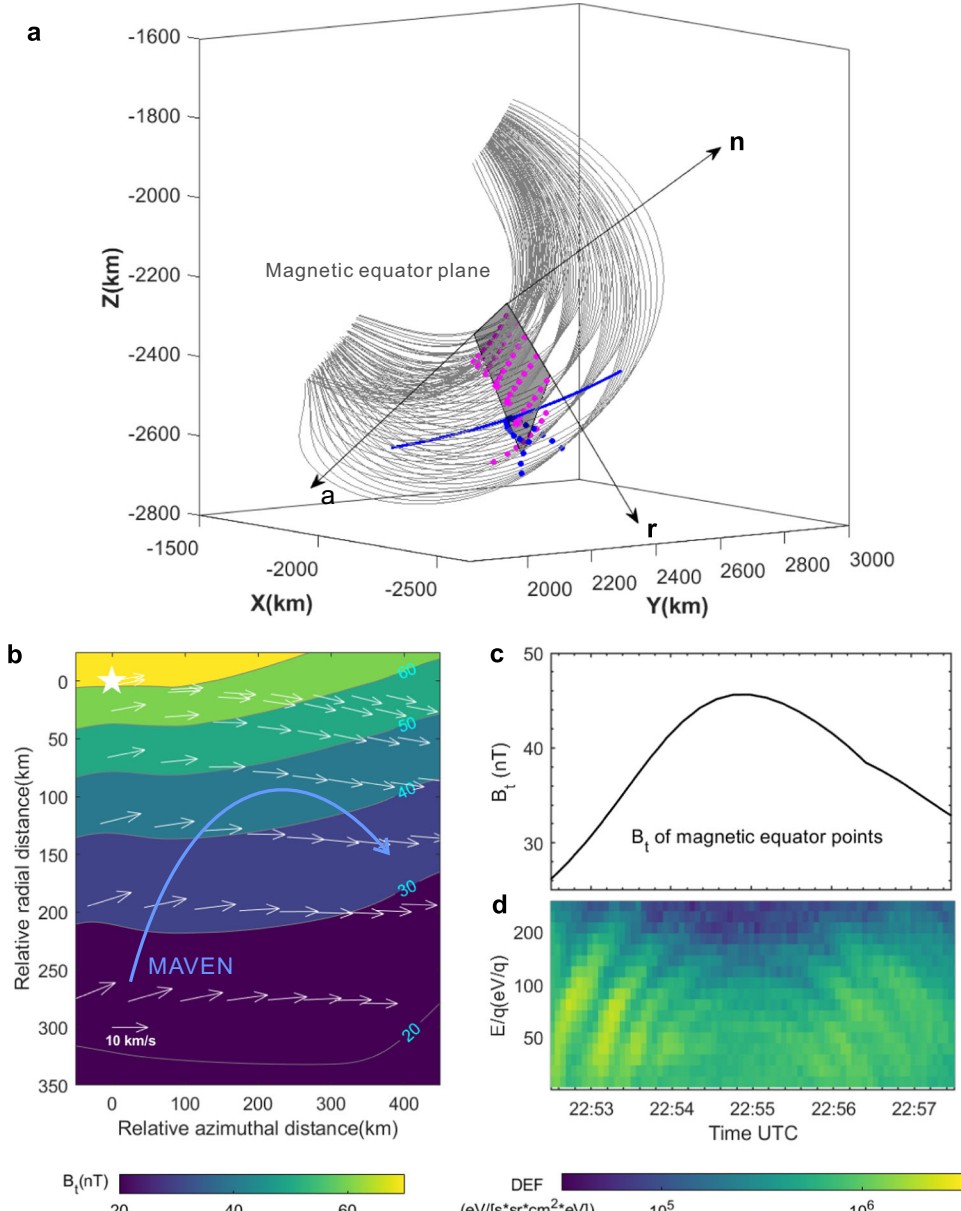

**Fig. 4 | Spatial separation of ions with different energies and the relative path of MAVEN on the magnetic equator plane of crustal fields. a** The distribution of magnetic equator points (magenta dots) of traced field lines (grey curves). Based on these magnetic equator points, we fit the magnetic equator plane (grey shaded region) that these magnetic equator points are almost located via the least-square method. The magnetic equator points of traced field lines along spacecraft's trajectory (blue curve) are marked by blue dots. The arrows dubbed as **a, r, n** denote the azimuthal direction, radial direction on the magnetic equator plane, and the normal direction of the magnetic equator plane, which consist of an orthogonal coordinate system {**n,a,r**}. **b** The distribution of the $B_t$ of magnetic equator points in the magnetic equator plane. The white arrows are the magnetic drift velocity of 100 eV H⁺ with 90° on these magnetic equator points (magenta dots in panel (**a**)). The blue curve with arrow represents the MAVEN path relative to the crustal fields. It should be noted that these magnetic equator points were not perfectly aligned with the magnetic equator plane, resulting in the magnetic drift velocity not being strictly orthogonal to $\nabla B_t$. The azimuthal and radial distance are the relative distance between each point to the origin point (the innermost magnetic equator point which is marked by the white star in the top left corner). **c** Time series of the $B_t$ of magnetic equator points of the traced magnetic field lines crossed by MAVEN. **d** The ion spectrum.

Another question is why the high-energy ions could drift deeper into the crustal fields. The total drift velocity of particles ($\mathbf{V_D}$) can be written as,

$$\mathbf{V_D} = \mathbf{V_{Mag.Drift}} + \mathbf{V_{Ele.Drift}}$$
$$= W\sin^2(\theta)/qB_t{}^3\mathbf{B} \times \nabla B_t + 2W\cos^2(\theta)/qB_t{}^4\mathbf{B} \times [(\mathbf{B} \cdot \nabla)\mathbf{B}] + \mathbf{E} \times \mathbf{B}/B_t{}^2 \quad (1)$$

where $W$,$q$ represents the energy, charge of ions, respectively. $\mathbf{B}$,$\mathbf{E}$,$B_t$ is the local magnetic fields, electric fields, and magnetic field strength, respectively. $\theta$ denotes the pitch angle of the particles. $\mathbf{V_{Mag.Drift}}$ represents the magnetic drift velocity, which consists of the gradient drift and the curvature drift, represented by the first and second term on the right-hand-side of Eq. (1). $\mathbf{V_{Ele.Drift}}$, the last term on the right-hand-side of Eq. (1), is the electric field drift velocity. In the case of Earth's inner magnetosphere with a dipole field pattern, the radial motion of ions is solely influenced by the electric field drift velocity since the magnetic drift velocity occurs exclusively along the azimuthal direction. Thus, the electric field drift motion plays a vital role in

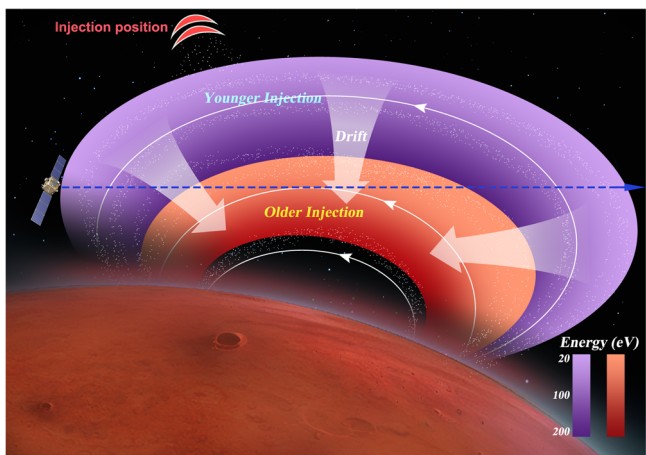

**Fig. 5 | Sketch of the wedge-like dispersions of ions within the Martian crustal fields.** The ions (white dots) with different energies are spatially separated due to their energy-dependent drift velocity (white thick arrows). The white thin curves with arrows represent the magnetic field lines of crustal fields. The ions of the older injection (younger) are illustrated by the red (violet) color, the shades of color represent the ion's energy. The dark (light) color represents the ion's energy is high (low). The red crescent-shaped area denotes the possible injection location. The blue dashed line with the arrow shows the MAVEN's path. The low-energy (high-energy) ions were located in the outer (inner) region of the crustal fields, leading to MAVEN recorded rising tones (falling tones) when moving towards the inner (outer) part of the crustal fields. The two injections resulted in the observations of two rising and falling tones. The older injected ions could reach the more interior region of the crustal fields compared with younger injected ions due to their longer drifting times. This would cause the spacecraft could only observe 20–100 eV dispersed ions for the older injection, while 20–200 eV dispersed ions for the younger injection.

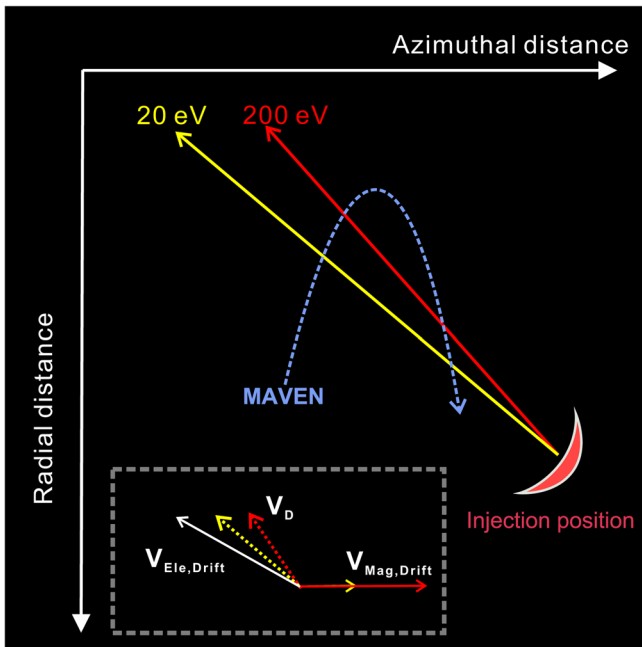

**Fig. 6 | A sketch of the possible drift path of dispersed ions.** The drift paths of 20 eV and 200 eV ions are represented by the long yellow and red arrows, respectively. The short solid and dashed lines with the yellow and red arrows indicate the magnetic drift velocity (purely along the azimuth direction) and total drift velocity of the 20 eV and 200 eV ions, respectively. The blue curve with an arrow depicts the relative spacecraft path. With the establishment of the electric field drift velocity (white arrow), the 200 eV ions would exhibit a greater propensity to drift deeper into the inner region compared to the 20 eV ions. This results in a radial spatial separation of ions with different energies.

forming the wedge-like dispersion structures, despite being independent of ion energy[22–27].

In our specific case, it is evident that the magnetic drift velocity on the magnetic equator plane of crustal fields is also primarily along the azimuthal direction and does not contribute to the radial transport of ions (as illustrated by the black arrows in Fig. 4b). This indicates that the radial transport of ions into the center of the crustal fields might also be facilitated by the electric field drift motion, akin to scenarios observed on Earth. Moreover, if we disregard the electric field drift, the total drift velocity of high-energy ions should be higher than the low-energy ions since $\mathbf{V}_{\mathbf{Mag,Drift}}$ is proportional to the ion's energy. Consequently, the observed rising tone structures indicate that MAVEN initially overtook the low-energy ions, followed by the high-energy ions. However, the magnetic drift speed of 50-200 eV ions ranges from 7-30 km/s, significantly exceeding the spacecraft's speed (about 3.8 km/s). This contradicts the aforementioned scenario and implies the involvement of the electric field drift. Nevertheless, further analysis remains constrained due to the limited information available on electric fields.

Here we employ the similar scenario forming the dispersed ions observed on Earth to elucidate our findings at Mars[22] (refer to Fig. 6). We establish an electric field drift velocity ($\mathbf{V}_{\mathbf{Ele,Drift}}$) that nearly balances the magnetic drift velocity ($\mathbf{V}_{\mathbf{Mag,Drift}}$) in azimuthal direction, with a limited radially inward component (refer to the white arrow in Fig. 6). Thus, the resulting total drift velocity ($\mathbf{V}_{\mathbf{D}}$) exhibits an inward tendency as the energy increases, since $\mathbf{V}_{\mathbf{Mag,Drift}}$ is proportional to the ion's energy. Consequently, the 200 eV ions exhibit a greater propensity to drift deeper into the inner region compared to the 20 eV ions. Then MAVEN would record one rising tone (falling tone) when moving towards the inner (outer) part of the crustal fields. In this scenario, the $\mathbf{V}_{\mathbf{Ele,Drift}}$ should nearly counterbalance the $\mathbf{V}_{\mathbf{Mag,Drift}}$ in the azimuthal direction. Taking into

consideration that the $|\mathbf{V}_{\mathbf{Mag,Drift}}|$ of 20-200 eV ions varies from 3-30 km/s and the average magnetic field strength is about 35 nT, the magnitude of electric field ($\mathbf{E} = -\mathbf{V}_{\mathbf{Ele,Drift}}\mathbf{B}$) contributing the azimuthal component of $\mathbf{V}_{\mathbf{Ele,Drift}}$ should be on the order of 0.1 mV/m. Furthermore, the observed rising tone structures indicate that, as MAVEN entered into the interior region of crustal fields, MAVEN should be faster than the dispersed ions along the radial direction (see Fig. 6). In other words, MAVEN's radial speed (about 1.1 km/s) should surpass the magnitude of $\mathbf{V}_{\mathbf{Ele,Drift}}$ along the radially inward direction. Consequently, the magnitude of the electric field ($\mathbf{E} = -\mathbf{V}_{\mathbf{Ele,Drift}}\mathbf{B}$) that contributes to the radial component of $\mathbf{V}_{\mathbf{Ele,Drift}}$ should be lower than 0.05 mV/m. Thus, the total magnitude of electric fields should be on the order of 0.1 mV/m, consistent with the latest simulation of the convective electric fields at Mars (0.1-1 mV/m)[49]. This value significantly exceeds the corotation electric field (0.01 mV/m) at this altitude (around 600 km), indicating that the potential existence of convective or impulsive electric fields.

These dispersed ions are anticipated to undergo adiabatic motion within the crustal fields. The gyroradius of 200 eV H$^+$ (representing the upper limit energy of dispersed ions) with a pitch angle of 90° spans a range of roughly 30 km to 100 km (Supplementary Fig. 6a). Meanwhile, the curvature radius of the field lines ranges from around 300 km to 1000 km (Supplementary Fig. 6b). Consequently, the adiabatic parameter ($\kappa$), defined as the square root of the ratio between the value of magnetic field curvature radius and the particle's gyroradius[50], consistently exceeds 2 (Supplementary Fig. 6c). This suggests that the dispersed ions predominantly undergo adiabatic motion, which aligns with our expectations.

In addition to the event discussed above, we have identified 16 additional wedge-like dispersion events within the crustal fields from the MAVEN dataset, spanning over a period of 5 years (refer to

Supplementary Table 1). The rare occurrence of drift dispersion structures might imply that their formation relies not only on the configuration of local magnetic fields, but also on the setup of electric fields that exhibit high variability at Mars. These observations highlight that both electrons and ions can become magnetized at Mars. Furthermore, for the ions, the Martian magnetosphere exhibits an intermediate case wherein both unmagnetized and magnetized ion behaviors are observable because of the wide range of strengths and spatial scales of crustal magnetic fields.

## Methods
### Instruments
Here we adopt ion data from the Solar Wind Ion Analyzer (SWIA)[32] and the Suprathermal, Thermal Ion Composition (STATIC) instrument[33], electron data from Solar Wind Electron Analyzer (SWEA)[35], and magnetic field measurements from the Magnetometer (M AG)[36], onboard MAVEN. The MAG is a fluxgate magnetometer that measures three-dimensional magnetic field vectors with frequencies of 32 Hz and 1 Hz. Here we use the 1 Hz data. SWIA can measure the ions with energy between 25 eV/q-25 keV/q at time resolution of 4 seconds during this event. The time resolution of full three-dimensional distribution of ions provided by SWIA is 8 seconds. Therefore, MAG and SWIA measurements can produce energy-resolved pitch angle distributions per 8 seconds. The angular resolution of SWIA for 20-200 eV ions is $22.5° \times 22.5°$, thereby we set that the bin size of pitch angle is $22.5°$, which is the upper limit of resolution of pitch angle. Here we use the c0 and c6 data of STATIC that provides the ions spectrum with energy between 0-30 keV/q at time resolution of 4 seconds. STATIC consists of a time-of-flight sensor which can measure the mass-per-charge of ions and then determine the specie of ions. The field-of-view of both SWIA and STATIC are $360° \times 90°$. During this event, SWEA could measure a full three-dimensional distribution of electrons with range from 3 eV/q to 46 KeV/q within 2 s. The field-of-view of SWEA is $360° \times 120°$, but about 8% field-of-view is blocked by the spacecraft.

### MSO coordinates
The coordinates utilized in this study are the Mars Solar Orbital (MSO) coordinates[51], where $\mathbf{X_{MSO}}$ points from Mars to the Sun, $\mathbf{Y_{MSO}}$ points opposite to the component of the orbital velocity perpendicular to $\mathbf{X_{MSO}}$, and $\mathbf{Z_{MSO}}$ completes the right-handed system.

### Magnetic Topology Analysis
The magnetic topology is categorized into seven types[38], (1) closed-to-day (C-D); (2) cross-terminator-closed (C-X); (3) closed-trapped (C-T); (4) closed-voids (C-V); (5) open-to-day (O-D); (6) open-to-night (O-N); (7) draped (DP). The closed-to-day type denotes field lines that have both foot-points on the dayside ionosphere, which can be identified by photoelectrons traveling in both parallel and antiparallel directions. The cross-terminator-closed type field line has one of the foot-points that ends in the dayside ionosphere, and the other one connected to the nightside ionosphere. For this type, both away-to-Mars and toward-to-Mars moving electrons are photoelectrons, but the flux of away-to-Mars moving electrons is higher. The closed-trapped type field lines are characterized by the electrons with pitch angle close to $90°$. If the trapped electrons are scattered into the loss cone, forming an electron void signature, which is identified by closed-voids type. For the open-to-day type, the field line has one of the foot-points that ends in the dayside ionosphere, which can be identified by the precipitation of solar wind electrons and an outflow of photoelectrons. The draped field lines are identified by solar wind electrons in both field-aligned directions, as well as by the absence of loss cones in the away directions. Therefore, (1), (2), (3), (4) typically represent the closed crustal fields. While (5), (6) represent the open field lines (OP). Supplementary

Fig. 2 shows that the field lines are mainly (3) or (4), where both are closed crustal fields.

### Normalized Pitch Angle Distribution
The normalized pitch Angle distribution are defined as the pitch angle distribution in units of differential energy flux as normalized by the average differential energy flux for each time step. That is $Norm, DEF = DEF/<DEF>$. $<DEF>$ represents the average differential energy flux for each time step.

### Tracing the magnetic field lines
We trace the magnetic field lines based on the superposed fields consisting of the crustal fields and induced fields with $[14.8, -0.93, -4.15]$nT. The crustal fields were derived from the latest spherical harmonic model of the crustal magnetic field[37].

We divide the spatial region within the latitude range of $-65°$ to $-55°$ and the longitude range of $175°$ to $190°$ into a grid with a spacing of $1°$ for both longitude and latitude. The altitude is then set at 500 km, resulting in 150 initial points, which are represented by $\mathbf{r_0}$, then the next position could be obtained by $\mathbf{r_1} = \mathbf{r_0} + \mathbf{b} \cdot \delta l$, where $\mathbf{b}$ is the unit magnetic field vector at $\mathbf{r_0}$, and $\delta l$ is the step length, here we set $\delta l = 1$km. Applying the same procedures, we could know the $i_{th}$ point of the field lines, $\mathbf{r_i}(i = 1,2,3..)$. We stop the tracing when the altitude reaches 200 km since the crustal field model could only well represent the crustal fields at altitudes above 120 km. After obtaining all points of the field lines, we regard the point with the smallest $B_t$ value as the magnetic equator point.

In addition, it should be noted that the gradients of the magnetic fields and the magnetic drift velocity are also calculated based on the superposed fields throughout this study.

### Fitting the magnetic equator plane
Assuming that the normal direction of a plane is $\mathbf{n} = A\mathbf{x} + B\mathbf{y} + C\mathbf{z}$ in the cartesian coordinate system $\{\mathbf{x,y,z}\}$. Then all the data points in this plane should satisfy the equation, $A \cdot P_x + B \cdot P_y + C \cdot P_z + D = 0$, where $\mathbf{P} = P_x\mathbf{x} + P_y\mathbf{y} + P_z\mathbf{z}$ represents the positions of data points, $D$ is constant. In other words, the data points on the plane would satisfy that, $\mathbf{P} \cdot \mathbf{n} + D = 0$. As the estimate of $\mathbf{n}$, the least-square method requires that identifies $(\{\mathbf{P}^{(m)} \cdot \mathbf{n}\}(m = 1,2,3...M))$ has the minimum variance, i.e., $\sigma^2 = \sum_{i=1}^{m} [|\mathbf{P}^{(m)} - <\mathbf{P}>| \cdot \mathbf{n}]^2/M$ reaches the minimum, where the average $<\mathbf{P}>$ is defined as $<\mathbf{P}> = \sum_{i=1}^{m} \mathbf{P}^{(m)}/M$. Furthermore, this minimization is subject to the normalization constraint $|\mathbf{n}|^2 = 1$. Then we could readily derive the $\mathbf{n}$ by the minimum variance analysis (MVA)[52], where $\mathbf{n}$ is the eigenvector corresponding to the minimum eigenvalue of the variance matrix $M_{ij} = <P_iP_j> - <P_i><P_j>$, where the subscripts $i,j = 1,2,3$ denote cartesian components along the $\{\mathbf{x,y,z}\}$ system. For our case, $\mathbf{n}$ is $(-0.7387, 0.5966, 0.3136)$. Then the azimuthal direction is deduced as $\mathbf{a} = (\mathbf{b} \times \mathbf{n})/(\mathbf{b} \times \mathbf{n})$, where $\mathbf{b}$ is the unit magnetic fields vector on the center of magnetic equator points, the radial direction ($\mathbf{r}$) completes the right-handed system. Here $\mathbf{r}$ is $(-0.1469, 0.3117, -0.9388)$, $\mathbf{a}$ is $(-0.6578, -0.7395, -0.1426)$.

## Data availability
All MAVEN data used in this paper are public. STATIC data can be found at https://lasp.colorado.edu/maven/sdc/public/data/sci/sta/l2/. MAG data can be found at https://lasp.colorado.edu/maven/sdc/public/data/sci/mag/l2/. SWIA data can be found at https://lasp.colorado.edu/maven/sdc/public/data/sci/swi/l2/. SWEA data can be found at https://lasp.colorado.edu/maven/sdc/public/data/sci/swe/l2/. The data of ion pitch angle distribution, magnetic topology, traced magnetic field lines, wave properties, and adiabatic parameters generated in this current study have been deposited in a Zenodo repository[53] (https://doi.org/10.5281/zenodo.8428367). The datasets generated during and/or analysed during the current study are available from the corresponding author upon request.

## Code availability

MAVEN data is analyzed and plotted mainly by the MAVEN Toolkit, which is publicly accessible from http://lasp.colorado.edu/maven/sdc/public/data/sdc/software/idl_toolkit/Toolkit_V2019-09-25_Public.zip. The pitch angle distribution of ions was computed by the IRFU-Matlab software, which is available by downloading from https://github.com/irfu/irfu-matlab. The codes for tracing the magnetic field lines and magnetic topology analysis are computed by the SPEDAS software[54], which is available at https://spedas.org/blog/. The code for the crustal fields model and the magnetic topology data utilized in this study is available at https://github.com/gaojiawei321/Mars_G110_model and the Zenodo repository at https://doi.org/10.5281/zenodo.8428367[53], respectively.

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

## Acknowledgements

This work is supported by the National Natural Science Foundation of China, grants No. 41922031 (Z. J. R), 41774188 (Z. J. R), the Strategic Priority Research Program of Chinese Academy of Sciences, Grant No. XDA17010201 (Y. W), XDB41000000 (Z. J. R), the Key Research Program of Chinese Academy of Sciences, Grant No. ZDBS-SSW-TLC00103 (Z. J. R), and the Key Research Program of the Institute of Geology & Geophysics, CAS, Grant No. IGGCAS- 201904 (Y. W), IGGCAS- 202102 (Z. J. R). We thank James P. McFadden, Li Li, Fei He, Zhonghua Yao, Wenya Li, Markus Fränz, and Ronan Modolo for their helpful discussion.

## Author contributions

C. Z, H. N, Y. E, M. Y and Z. J. R conceived this study. C. Z, H. N, Y. E, J. H, Y. H, M. Y, M. P, X. Z. Z, Y. X. S, J, H, C. J. Y, J. W. G and S. Z carried out the data analysis, interpretation and manuscript preparation. J. Z, C. F. D, Y. X. C, X. T. Y, S. F, X. Z. Z, Y. X. S, Y. F, M. P, J. W. G, M. H, Y. W, S. B carried out the manuscript revision and discussion. S. S. X performed the magnetic topology analysis and manuscript revision. J. W. G provided the code for crustal fields model. J. H contributed to the development and operation of the SWIA measurements, and manuscript revision. All authors contributed to the discussion and commented on the manuscript.

## Competing interests

The authors declare no competing interests.

## Inclusion & ethics statement

This research aligns with the Inclusion & ethical guidelines embraced by Nature Communications.
