## [Peer Review File · Nature Communications]

REVIEWER COMMENTS

Reviewer #1 (Remarks to the Author):

This manuscript presents very interesting observations and proposes a compelling explanation for the observations.

Overall, I think that the manuscript is well written and well organized.

I do feel, however, that the title is a bit over-the-top. The fact that the observations indicate energy-dependent drifts is not a property of intrinsic magnetospheres (though that is where they have been observed so far); it is a property of magnetic field configurations larger than the gyroradius of the particles. Such behavior has been hypothesized to occur in smaller-than-global magnetic field configurations elsewhere including the Moon and Mars. I think these observations are very interesting, and a first for Mars, but not necessarily surprising.

Additionally, I do have some relatively minor comments below.

Figure 4: I am having some trouble understanding Figure 4b. I guess in general I understand that it illustrates the magnitude of the magnetic field along the magnetic equatorial plane, but I don't understand the axes – particularly the vertical axis. What is the radial distance measured with respect to, and how does it go negative?

Figure 5 caption: I don't think that "polygon" is the best descriptor for the shaded shapes in Figure 5. Perhaps it is more of a crescent shape (polygon implies straight rather than curved edges).

Line 433: On this line SWIA should be SWEA since it has to do with electron measurements.

To what altitude are the magnetic field lines traced? High order spherical harmonic magnetic field models become unphysical at too low an altitude. The manuscript states that the magnetic field model represents the crustal fields well at altitudes above 120 km. Are the field lines traced lower than that altitude?

Additionally, although it is not explicitly stated, I assume that the gradient in the magnetic field is calculated using the magnetic field model plus the constant offset rather than just the model (this is mentioned for the field line tracing but not for the other magnetic field calculations).

Supplementary Figure 5 (and line 220): It is mentioned and shown that no proton cyclotron waves are present locally. However, Supplementary Figure 5c does appear to show some wave power at a slightly lower frequency on the order of 0.1 Hz. Are these waves significant and can they be identified?

Supplementary Table 1: I am curious as to why the solar wind dynamic pressure is included in this table. I don't recall dynamic pressure mentioned elsewhere in the manuscript. Additionally, it brings up questions as to how the dynamic pressure was calculated given that MAVEN is not in the solar wind during the time of observation. Personally, I would be interested in the possibility of variations in the upstream dynamic pressure during the observations; could dynamic pressure variations drive upstream waves (as mentioned on line 207)? But I wouldn't think that this information would be available given the spacecraft orbit.

Reviewer #2 (Remarks to the Author):

The paper presents interesting MAVEN plasma and magnetic field observations around closest approach, when the spacecraft traverse through a region of crustal magnetic fields. The authors report, for the first-time, observations of wedgelike dispersion signatures of protons, exhibiting butterfly-shaped distributions within the Martian crustal fields, a feature observed in intrinsic magnetospheres. The authors argue that the observed structures can be understood in terms of drifts motions, despite the relatively weak and large inhomogeneity of crustal magnetic fields, compared to planetary global intrinsic magnetic fields. The authors also report the identification of 16 additional wedge-like dispersion events within the crustal magnetic field of Mars, observed by MAVEN.

In my opinion, this manuscript presents new, original observations suggesting wedge-like dispersion signatures of protons at Mars. This is worthy of further investigation and opens the possibility for complementary studies involving magnetic reconnection and planetary ion escape, in connection with crustal magnetic fields. The main concern I have with this paper is that the authors do not sufficiently explain the observed signatures in terms of particle drifts within crustal magnetic fields. I think the manuscript would benefit from additional estimations of relevant parameters, based on

the properties observed by MAVEN and current crustal magnetic field models available in the literature.

Please find below my specific comments.

Line 98: How are the black dashed curves shown in panels 1a and 1b computed?

Line 109: Is there any physical explanation for a 20-30 s periodicity? What is the periodicity observed in the other 16 events the authors were able to identify?

Line 149: What is the interpretation for the additional [14.8, -0.93, -4.15] nT magnetic field?

Line 208-210: I think the observed periodicity should be explained in more detail.

Line 230-232: Why would the source be already anisotropic? Is it the case with all 16 additionally identified cases?

Line 362-364: According to the authors, the observations cannot be explained by magnetic drift, but by the electric drift velocity. Is there any evidence supporting this claim, in addition to the fact that the magnetic field drift would likely be azimuthal?

Line 377-379: If the estimated electric field is one order of magnitude larger than the corotation electric field, can the authors provide evidence in favor of the presence of convective or impulsive electric fields with the expected order of magnitude?

Lines 366-379: I think the manuscript would benefit from additional estimations. For instance, what is the proton Larmor radii range for the conditions observed in the analyzed event? How does it compare with the scale length over which the local (crustal) magnetic field varies? What is the expected electric field velocity drift that can explain the observations? How does it compare with the local proton velocity?

Reviewer #3 (Remarks to the Author):

Comments on “Discovery of Intrinsic Magnetospheric Ion Behaviour at Mars” by Zhang et al.

The manuscript describes observations of protons in Martian crustal magnetic fields that have dispersed energy signatures and oblique (or “butterfly”) pitch angle distributions that are asymmetric and correlated with the radial component of the crustal magnetic field. The manuscript discusses how the protons are injected into the crustal fields (reconnection), what causes their periodic occurrence in the crustal fields (periodic injections, heating, or variations in the source population), how the pitch angle distributions came to be asymmetric (scattering/drift of field-aligned/90-degree pitch angles), and how the energy dispersion signatures formed (higher energy ions drift deeper into the crustal fields).

Overall the phenomenon reported in the manuscript is novel for ions in the Martian magnetosphere (and any small-scale magnetosphere). I therefore believe that the community should be made aware of this work through publication, perhaps after some relatively minor comments below are considered. The main question in my mind is whether the work should be published in a high impact journal such as Nature Communications. Here, I am less certain, for three related reasons:

1. The signatures reported here for ions are similar in many ways (energy-time dispersion, bursty and periodic) to a previous report of observations of electrons in Mars crustal fields in 2016 by one of the manuscript’s co-authors, with similar interpretation about mechanisms. These similarities are not mentioned in the present manuscript at all, even though the paper (Harada et al.) is referenced. Though it is very interesting to see the same signatures in ions, what is learned overall from this manuscript about plasma behaviour in Mars crustal fields is therefore not particularly new.

2. The manuscript makes many references to “intrinsic magnetospheric behaviour” that the ion observations provide evidence for - saying in the opening paragraph that “...we report the discovery of intrinsic ion magnetospheric behavior at Mars...”. I am not fully convinced that this is a discovery, since there have been many previous reports of intrinsic magnetospheric behavior at Mars (e.g. aurora, magnetic reconnection). It may be that the emphasis in this phrase is on “ion” as opposed to “intrinsic magnetosphere”, but I find this less compelling since the community is already aware that the crustal fields behave as intrinsic magnetospheres in other regards, and there are published studies of ion observations near crustal fields from both Mars Express and MAVEN.

3. The manuscript mentions that only 17 examples of these observations have been found in 5 years of observations. This suggests that the evidence of intrinsic magnetospheric behaviour is the exception, and not the rule.

I note that the figures are informative and useful, and the paper is generally well-written.

My detailed comments follow:

L26-28: The wording of this sentence was awkward for me. I think I understand the intent: “The Martian magnetosphere is unusual and different from intrinsic magnetospheres”. But I might have worded it a little differently: “Mars lacks a global dynamo magnetic field, and instead possess small-scale crustal magnetic fields, making its magnetic environment fundamentally different from intrinsic magnetospheres like those of Earth or Saturn.”

L31: Clarify “butterfly-shaped pitch angle distributions”

L36-37: This last sentence of the paragraph is vague, and ultimately not revisited in the manuscript in any detail.

L43: suggest change to “...play a key role in the planet’s atmospheric...”

L48: suggest change to “...how do the crustal fields control...”

L49: re: “provide new insights” - this statement would be stronger if the kinds of new insights that are possible are described.

L52: suggest change to “mini-magnetospheres”

L53: The Harada et al. paper is referenced here, in a general statement about scaled-down intrinsic magnetospheres. But the electron signatures are so similar to the ion signatures that they should really be called out. Otherwise it feels like they are being deliberately minimized in the present work. (I'm not surmising this is the authors' intent. I'm just pointing out the impression it may give!)

L76-77: "can be regarded as key plasma evidence of intrinsic magnetosphere behaviour". This statement is setting up the argument for the present manuscript that dispersed ion structures are evidence of an intrinsic magnetosphere. It reads to me like a bit of an overstatement - that dispersed ion structures are the -only- evidence of intrinsic magnetosphere behaviour. However, there are many signatures of intrinsic magnetospheres, and several have been previously reported for Mars. And dispersed energy-time ion signatures have been previously reported in the Martian magnetosphere - just not near crustal fields.

L83: suggest changing "could" to "does"

L105-106: suggest change to "...traversed the nightside terminator..."

L133: add "the" before "energy"

L134: suggest changing "were" to "was"

L144: please provide a reference here in the text to the crustal field model, since there are many in the community.

L164: remove "meticulous"

L195-197: What is the evidence that magnetic reconnection causes the injections of ions? Are there other possibilities? This section has two sentences - one basically gives the question and the next gives the answer with no substantive justification. I suggest adding more discussion here.

L202-203: "the pitch angle...Br". I may have misunderstood this phrase, but think it could be reworded to be more clear. The specific pitch angle (e.g. 140 degrees) is not determined by Br. Instead, the asymmetry between parallel and antiparallel pitch angles is determined by Br.

L210: How could these three options be distinguished from each other?

L231: anisotropy → anisotropic

L269: coordinates → coordinate

L280-281: It seems like a lot of effort went into proving this statement that interior crustal field lines have higher field strength at the magnetic equator. This seems self-evident to me, without any demonstration required. Am I missing something? Second comment: The “MFL” and “MEP” acronyms are not standard in the literature, as far as I am aware. I understand why they were used - to avoid repetition in the discussion. But I ended up getting repeatedly lost while reading this paragraph and the several preceding paragraphs.

L303: suggest change to “According to the above results, the complete scenario of observed...”

L390-391: Are there peculiarities in the external conditions for the 17 identified events that could explain why they are only rarely observed?

L415-416: “crustal fields can locally provide a plasma environment similar to the planetary-scale intrinsic magnetospheres”. I agree with this statement, but don’t think this manuscript was the first to demonstrate this. Perhaps give other examples from the literature?

Response to the Reviewers

Reviewer #1 (Remarks to the Author):

This manuscript presents very interesting observations and proposes a compelling explanation for the observations.

Overall, I think that the manuscript is well written and well organized.

I do feel, however, that the title is a bit over-the-top. The fact that the observations indicate energy-dependent drifts is not a property of intrinsic magnetospheres (though that is where they have been observed so far); it is a property of magnetic field configurations larger than the gyroradius of the particles. Such behavior has been hypothesized to occur in smaller-than-global magnetic field configurations elsewhere including the Moon and Mars. I think these observations are very interesting, and a first for Mars, but not necessarily surprising.

Reply: We extend our gratitude to the reviewer for their valuable comment. Based on reviewer's suggestion, we have replaced the title by "**Discovery of typical intrinsic magnetospheric ion drift patterns at Mars**", so that the study content can be specifically reflected by the title for the better. The drift dispersion structures indeed indicate that the scale-sizes of gradient/curvature of the magnetic field configurations are larger than the gyroradius of the particles, but also requires stable configuration for both the magnetic field and electric field over the time scale that causes the dispersion. Since the scale-size magnetic field is small and complicated for the Mars, its field configurations may not be stable as the rotation of Mars.

It was found that the electron drift dispersion structures could occur within the small-scale crustal fields at Mars (Harada et al. 2015). Nevertheless, when considering ions (of ~100 eV) with larger gyroradius, drift dispersion structures of ions would not be anticipated at Mars, as indicated by Halekas et al. (2016), since the ions usually exhibit non-magnetized states characterized by the non-adiabatic motion of ions (Soobiah et al. 2019).

Therefore, our study reports the drift dispersion structures of ions at the first time despite MAVEN's continuous exploration of Mars for over 9 years. Our study also suggests that the Martian magnetosphere exhibits an intermediate case wherein both unmagnetized and magnetized ion behaviors are observable because of the wide range of strengths and spatial scales of induced and crustal magnetic fields.

Please see the revised Title, Abstract (lines 26-37), Introduction (lines 52-97), and Conclusion part (lines 450-454) of our paper.

Additionally, I do have some relatively minor comments below.

Figure 4: I am having some trouble understanding Figure 4b. I guess in general I understand that it illustrates the magnitude of the magnetic field along the magnetic

equatorial plane, but I don't understand the axes – particularly the vertical axis. What is the radial distance measured with respect to, and how does it go negative?

Reply: Thanks for this comment. In fact, we set the innermost magnetic equator point situated in the upper left corner as the origin point (we have marked it as the black star in the revised Figure 4). Consequently, both radial and azimuth distances are computed with respect to this origin point. We have clarified it in the paper. Please see the lines 278-279.

Figure 5 caption: I don't think that "polygon" is the best descriptor for the shaded shapes in Figure 5. Perhaps it is more of a crescent shape (polygon implies straight rather than curved edges).

Reply: We have replaced it by the crescent-shaped area. Please see the revised Figure 5.

Line 433: On this line SWIA should be SWEA since it has to do with electron measurements.

Reply: Thank you for catching the typo. We have revised it, please see line 469.

To what altitude are the magnetic field lines traced? High order spherical harmonic magnetic field models become unphysical at too low an altitude. The manuscript states that the magnetic field model represents the crustal fields well at altitudes above 120 km. Are the field lines traced lower than that altitude?

Reply: Here we traced all field lines starting at an altitude of 500 km, and we stopped tracing when the altitude reaches 200 km. We have described it in more detail, please see the subsection of "Tracing the magnetic field lines" of the section of "Method" (lines 499-507).

Additionally, although it is not explicitly stated, I assume that the gradient in the magnetic field is calculated using the magnetic field model plus the constant offset rather than just the model (this is mentioned for the field line tracing but not for the other magnetic field calculations).

Reply: Exactly. The gradient and magnetic drift velocity are all calculated based on the superposed fields, which are the magnetic field model plus the constant offset. We have emphasized in the subsection of "Tracing the magnetic field lines" of the section of "Method" (lines 508-509).

Supplementary Figure 5 (and line 220): It is mentioned and shown that no proton cyclotron waves are present locally. However, Supplementary Figure 5c does appear to show some wave power at a slightly lower frequency on the order of 0.1 Hz. Are these waves significant and can they be identified?

Reply: Thanks for this comment. We think these waves are not significant, since the variation of magnetic fields (32 Hz) (Supplementary Figure 5b) does not clearly show any wave signatures.

Supplementary Table 1: I am curious as to why the solar wind dynamic pressure is included in this table. I don't recall dynamic pressure mentioned elsewhere in the manuscript. Additionally, it brings up questions as to how the dynamic pressure was calculated given that MAVEN is not in the solar wind during the time of observation. Personally, I would be interested in the possibility of variations in the upstream dynamic pressure during the observations; could dynamic pressure variations drive upstream waves (as mentioned on line 207)? But I wouldn't think that this information would be available given the spacecraft orbit.

Reply: Thanks for this comment. We have deleted the information of solar wind dynamic pressure. The upstream conditions are considered to be the average values of the solar wind conditions in the 30 minutes preceding (subsequent to) the inbound (outbound) bow shock crossing for each orbit (see Halekas et al. 2017).

Reviewer #2 (Remarks to the Author):

The paper presents interesting MAVEN plasma and magnetic field observations around closest approach, when the spacecraft traverse through a region of crustal magnetic fields. The authors report, for the first-time, observations of wedgelike dispersion signatures of protons, exhibiting butterfly-shaped distributions within the Martian crustal fields, a feature observed in intrinsic magnetospheres. The authors argue that the observed structures can be understood in terms of drifts motions, despite the relatively weak and large inhomogeneity of crustal magnetic fields, compared to planetary global intrinsic magnetic fields. The authors also report the identification of 16 additional wedge-like dispersion events within the crustal magnetic field of Mars, observed by MAVEN.

In my opinion, this manuscript presents new, original observations suggesting wedge-like dispersion signatures of protons at Mars. This is worthy of further investigation and opens the possibility for complementary studies involving magnetic reconnection and planetary ion escape, in connection with crustal magnetic fields. The main concern I have with this paper is that the authors do not sufficiently explain the observed signatures in terms of particle drifts within crustal magnetic fields. I think the manuscript would benefit from additional estimations of relevant parameters, based on the properties observed by MAVEN and current crustal magnetic field models available in the literature.

We thank the reviewer for providing these insightful comments. Taking each comment into careful consideration, we have revised the paper, highlighting the changes in red text. Our paper has been improved based on your comments. We hope that our modifications have met the requirements of reviewer.

Please find below my specific comments.

Line 98: How are the black dashed curves shown in panels 1a and 1b computed?

Reply: The black dashed curves shown in panels 1a and 1b were manually fitted to accentuate the dispersed signatures, which does not impact the analysis presented in our paper. Although they may appear as curves, they are actually straight lines when viewed on a linear Y scale. However, for visual representation, they appear as curves when viewed on a logarithmic Y scale. Please see the line 108.

Line 109: Is there any physical explanation for a 20-30 s periodicity? What is the periodicity observed in the other 16 events the authors were able to identify?

Reply: As we discussed in the text, the bounce motion could not explain the periodicity of our event.

It is worth noting that periodical dynamics with periods of several tens of seconds have been commonly observed on Mars, as highlighted in previous studies by Zhang et al. (2021), Halekas et al. (2015, 2020), Grigoriev et al. (2007), and Collinson et al. (2018). These studies have suggested that such periodicity is attributed to upstream waves.

Based on this existing knowledge, we hypothesized that the periodicity observed in our event could also be attributed to these waves. It is widely accepted that these waves are primarily excited by planetary pickup ions or ions reflected by the bow shock (Dubinin & Fraenz, 2016; Halekas et al. 2020). Furthermore, these waves will exhibit a period approximately equal to the proton cyclotron period, which is on the order of 10 seconds within the upstream region.

While there are an additional 16 events demonstrating wedge-like dispersion structures, it's important to note that only a subset of these events (No. 4, 5, 9, 15 in Supplementary Table 1) display multiple dispersed structures. Remarkably, the periodicity observed in these events is also found to be on the order of 10 seconds, in alignment with our proposed mechanism. We have described it in more detail, please see the lines 223-228.

Line 149: What is the interpretation for the additional [14.8, -0.93, -4.15] nT magnetic field?

Reply: Based on the topology analysis and electron distributions, we believed that the magnetic field lines are mostly generated by the crustal fields. However, the crustal fields could interact with the induced fields, leading to the compression and distortion of the crustal fields. Therefore, we think the additional fields arise from the influence of the induced fields. Please see the lines 166-167.

Line 208-210: I think the observed periodicity should be explained in more detail.

Reply: Thank you, we have described it in more detail. Please see the lines 223-228.

Line 230-232: Why would the source be already anisotropic? Is it the case with all 16 additionally identified cases?

Reply: Given that the loss cone effect couldn't account for the difference between ions' pitch angles of 22.5° - 45° and 135° - 157.5° , we proposed that the source distribution itself might have been already anisotropic, leading to the observed anisotropy.

In fact, we found that each event exhibits a different pitch angle distribution. We will further investigate it in the future.

Line 362-364: According to the authors, the observations cannot be explained by magnetic drift, but by the electric drift velocity. Is there any evidence supporting this claim, in addition to the fact that the magnetic field drift would likely be azimuthal?

Reply: Thanks for this good comment. There are two indirect evidences pointing to the presence of electric fields. The first is the radial transport of ions, a phenomenon beyond the explanatory capacity of magnetic drift alone. The second is that the magnetic drift speed of 50-200 eV ions (7-30 km/s) significantly surpasses the spacecraft's velocity (~ 3.8 km/s). Thus, the spacecraft would exclusively detect the falling-tone (time-of-flight) dispersed structures since the magnetic drift speed is proportion to the energy, provided that we disregard the electric field drift.

Consequently, we think that the electric field drift (is energy independent and could contribute the radial transport) is involved for our specific case. Moreover, in addition to the case studies, if magnetic drift motion were indeed sufficient to elucidate the

formation of drift dispersion structures, one would anticipate a more prevalent occurrence of such structures, rather than the identification of only 17 distinct events, since the configuration of crustal fields is relatively stable while the electric fields are highly variable.

Nevertheless, the measurement of electric fields remains limited for MAVEN, depriving us of further substantiating evidence. Further observations and simulations might help us to shed light on it. We found that the words we wrote in the manuscript was too assertive, so we have also made modifications. Please see the lines 386-394.

Line 377-379: If the estimated electric field is one order of magnitude larger than the corotation electric field, can the authors provide evidence in favor of the presence of convective or impulsive electric fields with the expected order of magnitude?

Reply: We indirectly deduced the presence of convective or impulsive electric fields by comparing the magnitude of estimated electric field and the corotation electric field. But same to above, there are no confirmed direct evidence about the electric field. But the latest simulation about the global electric fields around Mars suggest that the convective electric field at ~500 km altitude ranges from 0.1-1 mV/m (Wang et al. 2023), align with our expectations. Please see the lines 402-416.

Lines 366-379: I think the manuscript would benefit from additional estimations. For instance, what is the proton Larmor radii range for the conditions observed in the analyzed event? How does it compare with the scale length over which the local (crustal) magnetic field varies? What is the expected electric field velocity drift that can explain the observations? How does it compare with the local proton velocity?

Reply: We have added the distribution of proton gyroradius, the curvature radius of field lines in the revised Supplementary Figure 6. We find that the gyroradius of 200 eV (the upper limit energy of dispersed ions) H^+ with pitch angle of 90° ranges from ~30 km to ~100km (Supplementary Figure 6a), while the curvature radius of the field lines varies from ~300 km to ~1000 km (Supplementary Figure 6b). Therefore, the gyroradius of protons are relatively smaller than the curvature radius.

The expected electric field velocity drift in the azimuthal direction should nearly counterbalance the magnetic drift velocity (3-30 km/s), while it should be smaller than the speed of MAVEN in the radial direction (~1.1 km/s) to lead to the observations of rising tones as MAVEN entered into the interior region of crustal fields.

The proton velocity is larger than the drift velocity since their field-aligned velocity ranges from 30-100 km/s. However, the field-aligned velocity will not affect the formation of the drift dispersion structures. Please see the lines 418-424.

Reviewer #3 (Remarks to the Author):

Comments on “Discovery of Intrinsic Magnetospheric Ion Behaviors at Mars” by Zhang et al.

The manuscript describes observations of protons in Martian crustal magnetic fields that have dispersed energy signatures and oblique (or “butterfly”) pitch angle distributions that are asymmetric and correlated with the radial component of the crustal magnetic field. The manuscript discusses how the protons are injected into the crustal fields (reconnection), what causes their periodic occurrence in the crustal fields (periodic injections, heating, or variations in the source population), how the pitch angle distributions came to be asymmetric (scattering/drift of field-aligned/90-degree pitch angles), and how the energy dispersion signatures formed (higher energy ions drift deeper into the crustal fields).

Overall the phenomenon reported in the manuscript is novel for ions in the Martian magnetosphere (and any small-scale magnetosphere). I therefore believe that the community should be made aware of this work through publication, perhaps after some relatively minor comments below are considered. The main question in my mind is whether the work should be published in a high impact journal such as Nature Communications. Here, I am less certain, for three related reasons:

Reply: We are grateful for the valuable suggestions and comments provided by the reviewer. We have considered your comments, and revised the paper on a point-by-point basis marked in red text. The paper has been enhanced significantly as a result of your insights. We trust that our revisions have successfully addressed the reviewer's requirements.

1. The signatures reported here for ions are similar in many ways (energy-time dispersion, bursty and periodic) to a previous report of observations of electrons in Mars crustal fields in 2016 by one of the manuscript's co-authors, with similar interpretation about mechanisms. These similarities are not mentioned in the present manuscript at all, even though the paper (Harada et al.) is referenced. Though it is very interesting to see the same signatures in ions, what is learned overall from this manuscript about plasma behaviour in Mars crustal fields is therefore not particularly new.

Reply: In fact, it is less expected that the drift dispersion of trapped ions on small-scale crustal fields would occur because of their large gyroradius, which is different to the case of electrons (Harada et al. 2015). Previous studies have only found the energy dispersed ions caused by the temporal effects (time-of-flight) and the ions were usually unmagnetized at Mars (Halekas et al. 2016; Soobiah et al. 2019; Zhang et al. 2021).

However, as opposed to the previous studies, our paper reported the observation of ion drift patterns typical for intrinsic magnetospheres within the Martian crustal fields at the first time despite MAVEN's continuous exploration of Mars for over 9 years. Our findings suggest that magnetized ion behavior commonly observed in intrinsic magnetospheres, does also occur at the crustal field regions of Mars. Furthermore, our

paper implies that the Martian magnetosphere embodies an intermediate case where both the unmagnetized and magnetized ion behaviors could be observed because of the wide range of strengths and spatial scales of induced and crustal magnetic fields. Thus, we think our paper presents a novel plasma signature at Mars, and might open the possibility for complementary studies involving magnetic reconnection and planetary ion escape, in connection with crustal magnetic fields. Please see the revised Title, Abstract (lines 26-37), Introduction (lines 52-97), and Conclusion part (lines 450-454) of our paper.

2. The manuscript makes many references to “intrinsic magnetospheric behaviour” that the ion observations provide evidence for - saying in the opening paragraph that “...we report the discovery of intrinsic ion magnetospheric behavior at Mars...”. I am not fully convinced that this is a discovery, since there have been many previous reports of intrinsic magnetospheric behavior at Mars (e.g. aurora, magnetic reconnection). It may be that the emphasis in this phrase is on “ion” as opposed to “intrinsic magnetosphere”, but I find this less compelling since the community is already aware that the crustal fields behave as intrinsic magnetospheres in other regards, and there are published studies of ion observations near crustal fields from both Mars Express and MAVEN.

Reply: We agree with the reviewer. In the revised manuscript, we have removed the terms of “intrinsic magnetospheric behavior”, but emphasized “ion behavior” and “Ion drift patterns” rather than “intrinsic magnetosphere”. We have revised the Abstract, Introduction and Discussion part of this paper.

For the importance of our paper, we think this paper updates our understanding of how the crustal fields controls the ion dynamics. Our work highlights that both electrons and ions can become magnetized, giving rise to drift dispersion structures on Mars. Furthermore, we suggest that the Martian magnetosphere exhibits an intermediate case wherein both unmagnetized and magnetized ion behaviors are observable because of the wide range of strengths and spatial scales of induced and crustal magnetic fields. As a result, we can further learn a rich variety of particle dynamics at Mars.

3. The manuscript mentions that only 17 examples of these observations have been found in 5 years of observations. This suggests that the evidence of intrinsic magnetospheric behavior is the exception, and not the rule.

Reply: We agree with the reviewer. That is why we think the development of dispersion structures of ions at Mars might hinge upon the setup of both magnetic fields and electric fields. It would be more common if the formation of dispersion structures of ions only depends on the magnetic field configuration. Please see the lines 429-431.

I note that the figures are informative and useful, and the paper is generally well-written.
Reply: Thanks for this positive comment.

My detailed comments follow:

L26-28: The wording of this sentence was awkward for me. I think I understand the intent: “The Martian magnetosphere is unusual and different from intrinsic magnetospheres”. But I might have worded it a little differently: “Mars lacks a global dynamo magnetic field, and instead possess small-scale crustal magnetic fields, making its magnetic environment fundamentally different from intrinsic magnetospheres like those of Earth or Saturn.”

Reply: We have revised it. Please see the lines 26-28.

L31: Clarify “butterfly-shaped pitch angle distributions”

Reply: We have clarified it. Please see the lines 31-32.

L36-37: This last sentence of the paragraph is vague, and ultimately not revisited in the manuscript in any detail.

Reply: Thanks for this comment. We have deleted it.

L43: suggest change to “...play a key role in the planet’s atmospheric...”

Reply: Revised, please see line 43.

L48: suggest change to “...how do the crustal fields control...”

Reply: Revised, please see line 48.

L49: re: “provide new insights” - this statement would be stronger if the kinds of new insights that are possible are described.

Reply: Deleted, please see line 49.

L52: suggest change to “mini-magnetospheres”

Reply: Revised, please see line 52.

L53: The Harada et al. paper is referenced here, in a general statement about scaled-down intrinsic magnetospheres. But the electron signatures are so similar to the ion signatures that they should really be called out. Otherwise it feels like they are being deliberately minimized in the present work. (I’m not surmising this is the authors’ intent. I’m just pointing out the impression it may give!)

Reply: We appreciate this comment very much. We have revised this manuscript and added the findings of Harada et al. (2015), please see the lines 83-85. We also have emphasized the importance and novelty of our paper. Please see the revised Introduction part.

L76-77: “can be regarded as key plasma evidence of intrinsic magnetosphere behaviour”. This statement is setting up the argument for the present manuscript that dispersed ion structures are evidence of an intrinsic magnetosphere. It reads to me like a bit of an overstatement - that dispersed ion structures are the -only- evidence of intrinsic magnetosphere behavior. However, there are many signatures of intrinsic magnetospheres, and several have been previously reported for Mars. And dispersed

energy-time ion signatures have been previously reported in the Martian magnetosphere - just not near crustal fields.

Reply: We agree with the reviewer. The drift dispersion structures of plasma indicate that the spatial scale of the electromagnetic environment is significantly larger than the plasma's gyroradius. We have revised it throughout this paper. Please see the lines 79-91.

Note that we so far have only found the energy dispersion of unmagnetized ions caused by temporal effect at Mars (Halekas et al. 2015; Zhang et al. 2021). The drift dispersion structures of ions have exclusively been documented within planetary-scale intrinsic magnetospheres with strong magnetic field strength, such as those of Earth and Saturn. Furthermore, their emergence within small-scale crustal fields is considerably unexpected since the ions are generally unmagnetized.

L83: suggest changing "could" to "does"

Reply: Changed, please see the line 96.

L105-106: suggest change to "...traversed the nightside terminator..."

Reply: Changed, please see the lines 99-100.

L133: add "the" before "energy"

Reply: Added, please see the line 144.

L134: suggest changing "were" to "was"

Reply: Changed, please see the line 145.

L144: please provide a reference here in the text to the crustal field model, since there are many in the community.

Reply: Added.

L164: remove "meticulous"

Reply: Removed.

L195-197: What is the evidence that magnetic reconnection causes the injections of ions? Are there other possibilities? This section has two sentences - one basically gives the question and the next gives the answer with no substantive justification. I suggest adding more discussion here.

Reply: We have discussed two possible mechanisms that result in the ions injection: magnetic reconnection and the non-adiabatic effect. Demonstrating both scenarios is challenging due to limited observations. Further studies might address it. We have expanded our discussion on these aspects. Please see the lines 204-214.

L202-203: "the pitch angle...Br". I may have misunderstood this phrase, but think it could be reworded to be more clear. The specific pitch angle (e.g. 140 degrees) is not determined by Br. Instead, the asymmetry between parallel and antiparallel pitch angles

is determined by Br.

Reply: Deleted.

L210: How could these three options be distinguished from each other?

Reply: If the injection position is stationary, the periodic injections and variations in the source population should be identical; otherwise, they would be different. Nonetheless, determining the stationarity of the injection position proves to be challenging.

L231: anisotropy —> anisotropic

Reply: Revised, please see the line 249.

L269: coordinates —> coordinate

Reply: Revised, please see the line 290.

L280-281: It seems like a lot of effort went into proving this statement that interior crustal field lines have higher field strength at the magnetic equator. This seems self-evident to me, without any demonstration required. Am I missing something? Second comment: The “MFL” and “MEP” acronyms are not standard in the literature, as far as I am aware. I understand why they were used - to avoid repetition in the discussion. But I ended up getting repeatedly lost while reading this paragraph and the several preceding paragraphs.

Reply: The crustal magnetic fields exhibit greater complexity and irregularity compared to the dipole fields. Proving that interior crustal field lines have higher field strength at the magnetic equator before the further analysis is better, despite the fact that the morphology of localized crustal fields appears dipole-like in our specific case.

The “MFL” and “MEP” acronyms are removed throughout this paper.

L303: suggest change to “According to the above results, the complete scenario of observed...”

Reply: Changed, please see the line 328.

L390-391: Are there peculiarities in the external conditions for the 17 identified events that could explain why they are only rarely observed?

Reply: We have not identified any distinct peculiarities in the external conditions for the 17 events. We think the development of dispersion structures of ions at Mars might hinge upon the setup of both magnetic fields and electric fields. Please see the lines 429-431. Nevertheless, it is important to note that the absence of concurrent measurements for both solar wind and crustal fields could impose limitations on our further research.

L415-416: “crustal fields can locally provide a plasma environment similar to the planetary-scale intrinsic magnetospheres”. I agree with this statement, but don’t think this manuscript was the first to demonstrate this. Perhaps give other examples from the literature?

Reply: Thanks for this comment. We agree with the reviewer. Our paper actually reports the wedge-like dispersion structures of ions at Mars at the first time, which are previously only reported in intrinsic magnetospheres. We have revised the paper, please see the lines 434-435, 450-454.

Reference:

- Collinson, G. (2018). Solar Wind Induced Waves in the Skies of Mars: Ionospheric Compression, Energization, and Escape Resulting From the Impact of Ultralow Frequency Magnetosonic Waves Generated Upstream of the Martian Bow Shock. *Journal of Geophysical Research: Space Physics*, 123(5), 4129-4149. <https://doi.org/10.1029/2018JA025414>
- Dubinin, E., Fraenz, M., (2016). Waves at Venus and Mars. In: Keiling, A., V. Nakariakov, D.H.L. (Eds.), *Low-frequency Waves in Space Plasmas*, vol. 103. AGU, pp. 343–364.
- Grigoriev, A., Futaana, Y., Barabash, S., & Fedorov, A. (2007). Observations of the Martian Subsolar ENA Jet Oscillations. *Space Science Reviews*, 126(1-4), 299-313. <https://doi.org/10.1007/s11214-006-9121-y>
- Halekas, J. S., et al. (2015), Time-dispersed ion signatures observed in the Martian magnetosphere by MAVEN, *Geophys. Res. Lett.*, 42, 8910–8916, doi:10.1002/2015GL064781.
- Halekas, J. S., et al. (2017), Structure, dynamics, and seasonal variability of the Mars solar wind interaction: MAVEN Solar Wind Ion Analyzer in-flight performance and science results, *J. Geophys. Res. Space Physics*, 122, 547–578, doi:10.1002/2016JA023167.
- Halekas, J. S., Ruhunusiri, S., Vaisberg, O. L., Harada, Y., Espley, J. R., Mitchell, D. L., Mazelle, C., Romanelli, N., DiBraccio, G. A., & Brain, D. A. (2020). Properties of Plasma Waves Observed Upstream From Mars. *Journal of Geophysical Research: Space Physics*, 125(9). <https://doi.org/10.1029/2020ja028221>
- Harada, Y., et al. (2016), MAVEN observations of energy-time dispersed electron signatures in Martian crustal magnetic fields, *Geophysical Research Letters*, 43, 939–944, doi:10.1002/2015GL067040.
- Soobiah, Y. I. J., Espley, J. R., Connerney, J. E. P., et al.. (2019). MAVEN Case Studies of Plasma Dynamics in Low-Altitude Crustal Magnetic Field at Mars 1: Dayside Ion Spikes Associated With Radial Crustal Magnetic Fields. *Journal of Geophysical Research: Space Physics*, 124(2), 1239-1261. <https://doi.org/10.1029/2018ja025569>
- Wang, X.-D., Fatemi, S., Nilsson, H., Futaana, Y., Holmström, M., & Barabash, S. (2023). Solar wind interaction with Mars: electric field morphology and source terms. *Monthly Notices of the Royal Astronomical Society*, 521(3), 3597-3607. <https://doi.org/10.1093/mnras/stad247>
- Zhang, C., Rong, Z., Nilsson, H., Klinger, L., Xu, S., Futaana, Y., Wei, Y., Zhong, J., Fränz, M., Li, K., Zhang, H., Fan, K., Wang, L., Holmström, M., Ge, Y., & Cui,

J. (2021). MAVEN Observations of Periodic Low-altitude Plasma Clouds at Mars. *The Astrophysical Journal Letters*, 922(2). <https://doi.org/10.3847/2041-8213/ac3a7d>

REVIEWERS' COMMENTS

Reviewer #1 (Remarks to the Author):

I am very sorry for my late response. The response to the reviewers and the revised manuscript have addressed most of my major concerns.

The only question that popped up to me is that of the speed of the MAVEN spacecraft. On line 392, the speed is quoted at “~3.8 km/s” while on line 410 it is “~1.3 km/s”. Is this second speed just the radial component? It is not clear to me.

Aside from that, I personally think the manuscript presents interesting and convincing analyses.

Reviewer #2 (Remarks to the Author):

I thank the authors for addressing my comments. I suggest publication in Nature Communications.

Reviewer #3 (Remarks to the Author):

I thank the authors for their careful responses to the reviewer comments, and their corresponding revisions to the manuscript. I am happy with their choice to recast the importance of the results in terms of observing ion drift patterns, and the Mars crustal magnetic fields as an intermediate case that can host both magnetized and unmagnetized ions.

We are thankful for the positive comments provided by the reviewers.

Reviewer #1 (Remarks to the Author):

I am very sorry for my late response. The response to the reviewers and the revised manuscript have addressed most of my major concerns.

The only question that popped up to me is that of the speed of the MAVEN spacecraft. On line 392, the speed is quoted at " ~ 3.8 km/s" while on line 410 it is " ~ 1.3 km/s". Is this second speed just the radial component? It is not clear to me.

Aside from that, I personally think the manuscript presents interesting and convincing analyses.

Reply: Thank you for this comment. It is right, the 3.8 km/s is the total speed, while the 1.3 km/s is the radial speed. I have added it in the text, please see the line 412.

Reviewer #2 (Remarks to the Author):

I thank the authors for addressing my comments. I suggest publication in Nature Communications.

Reviewer #3 (Remarks to the Author):

I thank the authors for their careful responses to the reviewer comments, and their corresponding revisions to the manuscript. I am happy with their choice to recast the importance of the results in terms of observing ion drift patterns, and the Mars crustal magnetic fields as an intermediate case that can host both magnetized and unmagnetized ions.